# Aversive stimuli drive hypothalamus-to-habenula excitation to promote escape behavior

Salvatore Lecca[1,2], Frank Julius Meye[1,3], Massimo Trusel[1,2], Anna Tchenio[1,2], Julia Harris[4], Martin Karl Schwarz[5], Denis Burdakov[4], Francois Georges[6,7], Manuel Mameli[1,2]*

[1]Institut du Fer à Moulin, Inserm UMR-S 839, Paris, France; [2]Department of Fundamental Neuroscience, The University of Lausanne, Lausanne, Switzerland; [3]Department Translational Neuroscience, Brain Center Rudolf Magnus, University Medical Center Utrecht, Utrecht, Netherlands; [4]The Francis Crick Institute, London, United Kingdom; [5]Clinic for Epilepsy Life and Brain Center, University Clinic of Bonn, Bonn, Germany; [6]Université de Bordeaux, Neurodegeneratives Diseases Institute, Bordeaux, France; [7]Centre National de la Recherche Scientifique, Neurodegeneratives Diseases Institute, Bordeaux, France

**Abstract** A sudden aversive event produces escape behaviors, an innate response essential for survival in virtually all-animal species. Nuclei including the lateral habenula (LHb), the lateral hypothalamus (LH), and the midbrain are not only reciprocally connected, but also respond to negative events contributing to goal-directed behaviors. However, whether aversion encoding requires these neural circuits to ultimately prompt escape behaviors remains unclear. We observe that aversive stimuli, including foot-shocks, excite LHb neurons and promote escape behaviors in mice. The foot-shock-driven excitation within the LHb requires glutamatergic signaling from the LH, but not from the midbrain. This hypothalamic excitatory projection predominates over LHb neurons monosynaptically innervating aversion-encoding midbrain GABA cells. Finally, the selective chemogenetic silencing of the LH-to-LHb pathway impairs aversion-driven escape behaviors. These findings unveil a habenular neurocircuitry devoted to encode external threats and the consequent escape; a process that, if disrupted, may compromise the animal's survival.

DOI: https://doi.org/10.7554/eLife.30697.001

*For correspondence: manuel.mameli@unil.ch

## Introduction

Unexpected threats drive escape behavior, an innate (non-conditioned) response crucial for survival (*Fadok et al., 2017*; *Mongeau et al., 2003*); yet the required neurocircuitry guiding such behavioral response is unclear. The lateral habenula (LHb), through the computing of its incoming inputs (i.e. basal ganglia, hypothalamus, midbrain), encodes disappointment and the negative value of a stimulus (*Hong and Hikosaka, 2008*; *Knowland et al., 2017*; *Matsumoto and Hikosaka, 2007*; *Root et al., 2014b*; *Shabel et al., 2012*; *Stamatakis et al., 2016*; *Stephenson-Jones et al., 2016*; *Wang et al., 2017*). Glutamatergic LHb neurons directly innervate midbrain GABA cells, and indirectly inhibit dopamine cells, two neuronal populations known to respond to aversive stimuli (*Brischoux et al., 2009*; *Stamatakis and Stuber, 2012*; *Tan et al., 2012*). Notably, a subset of habenular axons also innervates aversion-encoding cortex-projecting dopamine neurons (*Balcita-Pedicino et al., 2011*; *Lammel et al., 2012*). These findings support a scenario in which the LHb is a brain center encoding aversion mostly through the inhibition of the reward system (*Ji and Shepard,*

*2007*). However, the nature of the habenular inputs underlying the processing of aversive stimuli, and their subsequent behavioral relevance remain unclear.

The lateral hypothalamus (LH) and the medial portion of the ventral tegmental area (mVTA), innervate the LHb (*Herkenham and Nauta, 1977*). Both the LH and the mVTA contain heterogeneous neuronal populations, but afferents to the LHb are mostly excitatory (*Root et al., 2014b*; *Stamatakis et al., 2016*). Notably, such LH and mVTA populations respond to negative events and contribute to shape aversive behaviors including escape (*Morales and Margolis, 2017*; *Herkenham and Nauta, 1977*; *Matsumoto and Hikosaka, 2007*; *González et al., 2016*; *Wang et al., 2015*). However, whether these projections instruct habenular neurons to process aversive stimuli and to ultimately orchestrate escape behaviors is unknown.

To test this hypothesis, we use a combination of electrophysiology with chemo- and optogenetics to show that the hypothalamic-to-habenula inputs, but not the mVTA-to-habenula projection, mediate foot-shock-driven glutamate-dependent excitation of the LHb. We demonstrate that such hypothalamic-habenular circuit underlies aversion-driven escape, unraveling a behavioral consequence of aversion processing within the LHb.

## Results

### Aversive stimuli lead to glutamate-driven excitation of lateral habenula

To model aversive escape behaviors, we placed mice in a two-compartment chamber and delivered a series of unpredicted foot-shocks (30/session). Across trials, mice rapidly escaped to the compartment opposite to the one of shock delivery (*Figure 1A*). To explore habenula's contribution to such behavioral response, we LHb-targeted a viral construct coding for the calcium sensor GCamp6f (rAAV-hSyn-GCamp6f-eGFP) (*Figure 1A*). Positioning a multimode optical probe above the LHb allowed fluorescence transients detection, which represent bouts of neuronal activity (*Cui et al., 2014*). Foot-shocks (0.3 mA), and aversive air-puffs (painless stimulus), evoked escape behavior (i.e. phasic increase in locomotion) along with time-locked fluorescent transients within the LHb (*Figure 1A*, *Figure 1—figure supplement 1A–B*). Notably, a progressive increase in locomotion in absence of foot-shock, artificially promoted using the rotarod, did not modify the fluorescence signal measured from the LHb (*Figure 1—figure supplement 1C*). Therefore, simultaneous monitoring of fluorescence and behavior suggests a relationship between LHb activity and escape (*Lawson et al., 2014*; *Matsumoto and Hikosaka, 2007*).

To examine aversive stimuli-mediated single-cell LHb dynamics, we analyzed single-unit activities in anesthetized mice. Foot-shocks (hindpaw delivery) led to fast (~60 ms) and phasic increase (~100 ms) in action potentials in ~50% of neurons recorded across the LHb (*Figure 1A–D*, *Figure 1—figure supplement 2A–D*). Local infusion of glutamate receptors antagonists (APV and NBQX) diminished foot-shock-driven excitation of LHb neurons, without altering baseline firing, indicating the glutamatergic nature of the response (*Figure 1E*, *Figure 1F*, *Figure 1—figure supplement 2E–F*). Altogether, aversive stimuli generate a rapid glutamate-dependent excitation of a subset of LHb neurons.

### Control of aversion processing by hypothalamic-habenular projections

What exact circuit architecture governs aversion-driven excitation of LHb? Glutamatergic projections from the medial ventral tegmentum (mVTA) in the midbrain and the lateral hypothalamus (LH) innervate almost-entirely the LHb (*Stamatakis et al., 2013*; *Herkenham and Nauta, 1977*; *Barker et al., 2016*; *Kunwar et al., 2015*; *Root et al., 2014a*; *2014b*; *Stamatakis et al., 2016*), indicating their potential implication in conveying aversive information onto habenular neurons.

Firstly, we examined mVTA and LH inputs synaptic nature onto LHb. In different sets of mice, we virally expressed either in mVTA or LH the excitatory opsin from *Chloromonas oogama* (rAAV-hSyn-CoChR-eGFP) (*Klapoetke et al., 2014*). Optical stimulation (1 ms; 470 nm) of CoChR-expressing mVTA or LH terminals (VTA→LHb and LH→LHb respectively), in acute slices, evoked inward glutamatergic currents onto recorded neurons (−50 mV) across the LHb (*Figure 2A–B*). Moreover, in behaving mice, optogenetic activation of either VTA→LHb or LH→LHb was sufficient to produce place aversion (*Figure 2—figure supplement 1A–B*) (*Root et al., 2014a*; *Stamatakis et al., 2016*). This supports the notion that VTA→LHb and/or LH→LHb can underlie aversive processing in the

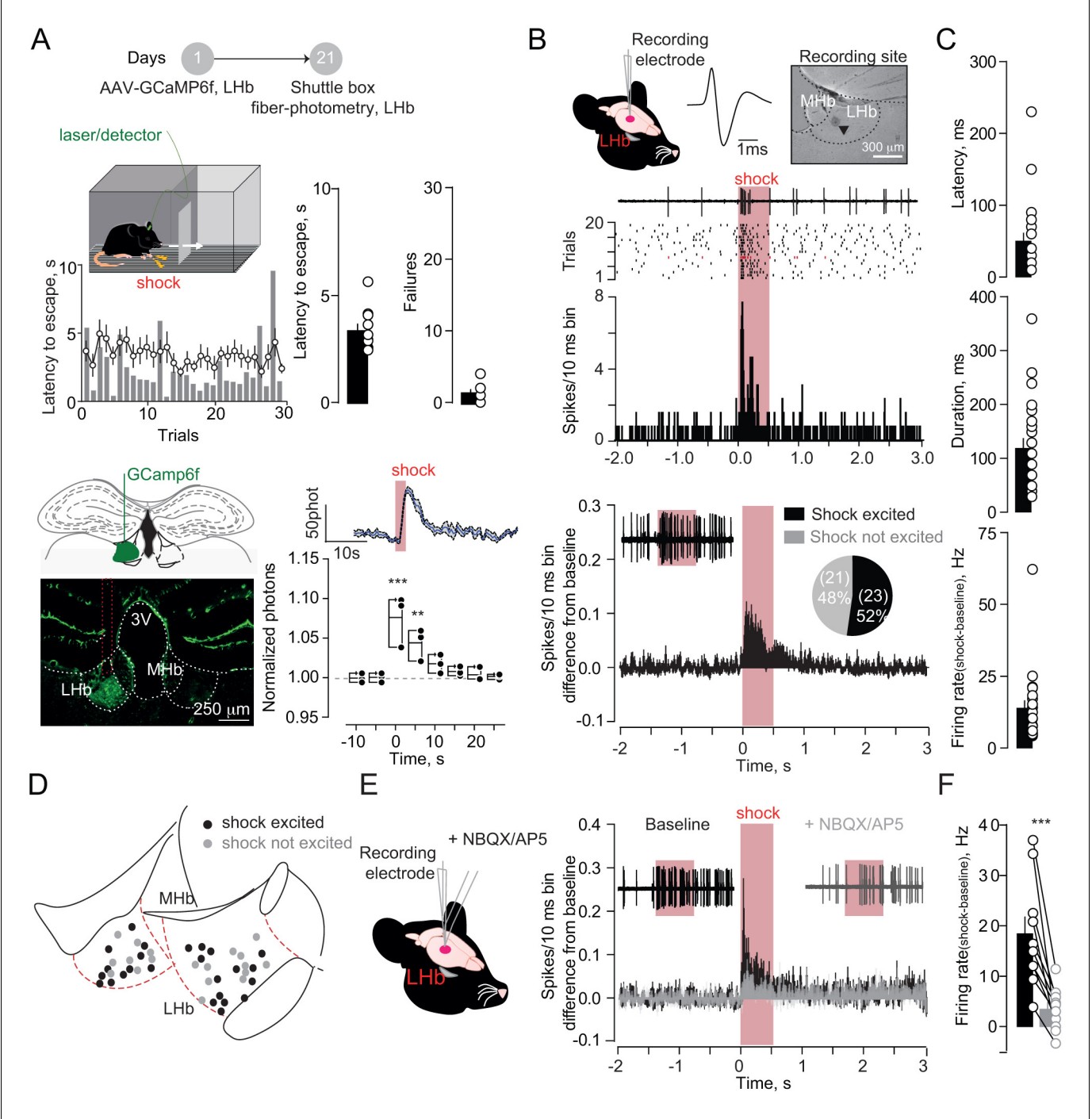

**Figure 1.** Foot-shocks promote escape behavior and glutamate-dependent phasic excitation of LHb. (**A**) Experimental timeline, representative histogram and bar graphs reporting foot-shock (Fs)-driven mouse behavior (latency to escape and failures) (N = 10). Bottom, GCamp6f expression in LHb and Fs-mediated Ca$^{2+}$ transients (N = 3, one way ANOVA RM, $F = 14.24$, ***$p<0.0001$). (**B**) Spike waveform and recording location of a LHb neuron (arrow, pontamine sky blue dye). Representative trace, raster plot and peristimulus time histogram (PSTH) for Fs-evoked excitation (Fs, 0.5 s, 3mA, ISI: 5 s). Bottom. PSTH reporting average spike counting and pie-chart for Fs-excited LHb neurons ((Firing rate/10 ms-average 2 s pre-shock)/Total trials) (N/n = 7/44). (**C**) Analysis of Fs excitation in LHb neurons. (**D**) Territorial distribution of Fs-excited and not-excited neurons. (**E**) PSTHs reporting Fs responses before/after local infusion of NBQX/AP5. (**F**) Bar graph and scatter plot for shock-driven activity before/after NBQX/AP5 (N/n = 6/10, paired t-test, $t = 5.05$; ***$p=0.0007$). Results are reported as mean ±S.E.M. N = mice; n = cells. 3V, 3$^{rd}$ ventricle, MHb, medial habenula. See *Figure 1—source data 1*.

DOI: https://doi.org/10.7554/eLife.30697.002

*Figure 1 continued on next page*

*Figure 1 continued*

The following source data and figure supplements are available for figure 1:

**Source data 1.**

DOI: https://doi.org/10.7554/eLife.30697.005

**Figure supplement 1.** Aversive stimuli and locomotion differentially affect LHb neuronal activity.

DOI: https://doi.org/10.7554/eLife.30697.003

**Figure 1—supplement figure 1—source data 1.**

DOI: https://doi.org/10.7554/eLife.30697.006

**Figure supplement 2.** Firing properties and pharmacology of foot-shock responses in LHb neurons of anesthetized mice.

DOI: https://doi.org/10.7554/eLife.30697.004

**Figure 1—supplement figure 2—source data 1.**

DOI: https://doi.org/10.7554/eLife.30697.007

LHb and escape behaviors. We then tested whether both VTA→LHb and LH→LHb projections are, not only sufficient, but also necessary.

We targeted the mVTA or the LH with a viral construct coding for the inhibitory designer receptors exclusively-activated by designer drugs, DREADDi (rAAV8-hSyn-hM4Di-mCherry), to silence VTA→LHb LHb or LH→LHb inputs (*Zhu and Roth, 2014*) (*Figure 2C* and *Figure 2D*). To ascertain the efficacy of DREADDi in suppressing input-specific presynaptic release, we co-infused two viral vectors coding for CoChR and DREADDi within the LH (*Figure 2—figure supplement 2A*). Indeed, in acute slices, application of the DREADDi agonist clozapine-N-oxide (CNO) promptly reduced LH→LHb light-evoked excitatory postsynaptic currents and increased the paired-pulse ratios (*Figure 2—figure supplement 2B–C*). When recording LHb neuronal activity in vivo in anesthetized animals, local CNO infusion in mice expressing DREADDi in the LH→LHb, but not in the VTA→LHb, significantly reduced foot-shock-mediated excitation without altering baseline activity (*Figure 2C* and *Figure 2D*, *Figure 2—figure supplement 2D*). Consistently, retrogradely-labeled and optically-tagged LHb-projecting LH neurons were foot-shock responsive (*Figure 2—figure supplement 3A–E*). Altogether, the LH→LHb, but not VTA→LHb projection, underlies foot-shock excitation of LHb.

## Targeted and cell-type trans-synaptic tracing in the lateral habenula

Neurons of the LHb send their projections onto midbrain GABA, dopamine (DA) and serotonin (5HT) cells thereby engaging monoaminergic nuclei, which also contribute to aversion processing (*Auerbach et al., 1985*; *Brischoux et al., 2009*; *Lammel et al., 2012*; *Pollak Dorocic et al., 2014*; *Stamatakis and Stuber, 2012*; *Tan et al., 2012*). We sought to decipher the architecture of [LH]-LHb projections to midbrain GABA, DA or 5HT cells using *slc30a1-Cre* (VGat-Cre), *Pitx3-Cre* and *Slc6a4-Cre* (Sert-Cre) mice in combination with a retrograde cell-type-specific monosynaptic labeling strategy (RABVΔG-(EnvA)-eGFP). We employed this along with the activation of channelrhodopsin-2 (rAAV-CAG-ChR2(H134R)-mCherry)-expressing LH terminals to probe the strength of the following synapses: LH→LHb-to-GABA, LH→LHb-to-DA, and LH→LHb-to-5HT (*Figure 3A*). LHb-to-GABA cells were mostly located in the lateral portion of the LHb, in contrast to those projecting to DA- and 5HT neurons that were medially-located (*Figure 3A* and *Figure 3—figure supplement 1A* [*Meye et al., 2016*]). When recording from these output-identified LHb neurons in acute slices, we found that LH→LHb opto-stimulation led to inward (−60 mV, I-AMPA) and outward current responses (+10 mV, I-GABA) (*Herrera et al., 2016*; *Stamatakis et al., 2016*). LH synapses onto LHb-to-GABA and LHb-to-DA had high degree of connectivity (~90% and ~80% respectively) in contrast to those projecting to LHb-to-5HT neurons (~50%) (*Figure 3B* and *Figure 3—figure supplement 1A–B*). We then computed the I-GABA/I-AMPA ratios as a measure for the dominant synaptic component (*Meye et al., 2016*). The LH→LHb-to-GABA neurons presented lower I-GABA/I-AMPA ratios, and larger maximal I-AMPA compared to other LHb cell targets (*Figure 3B*), suggesting that LH excitation predominates over LHb neurons synapsing onto downstream midbrain GABA cells.

## Hypothalamic-habenular projections guide escape behaviors in mice

If LH inputs instruct LHb-to-midbrain projections to orchestrate aversion-driven escape behaviors, our prediction is that impairing their function may be detrimental for animal's ability to cope with a

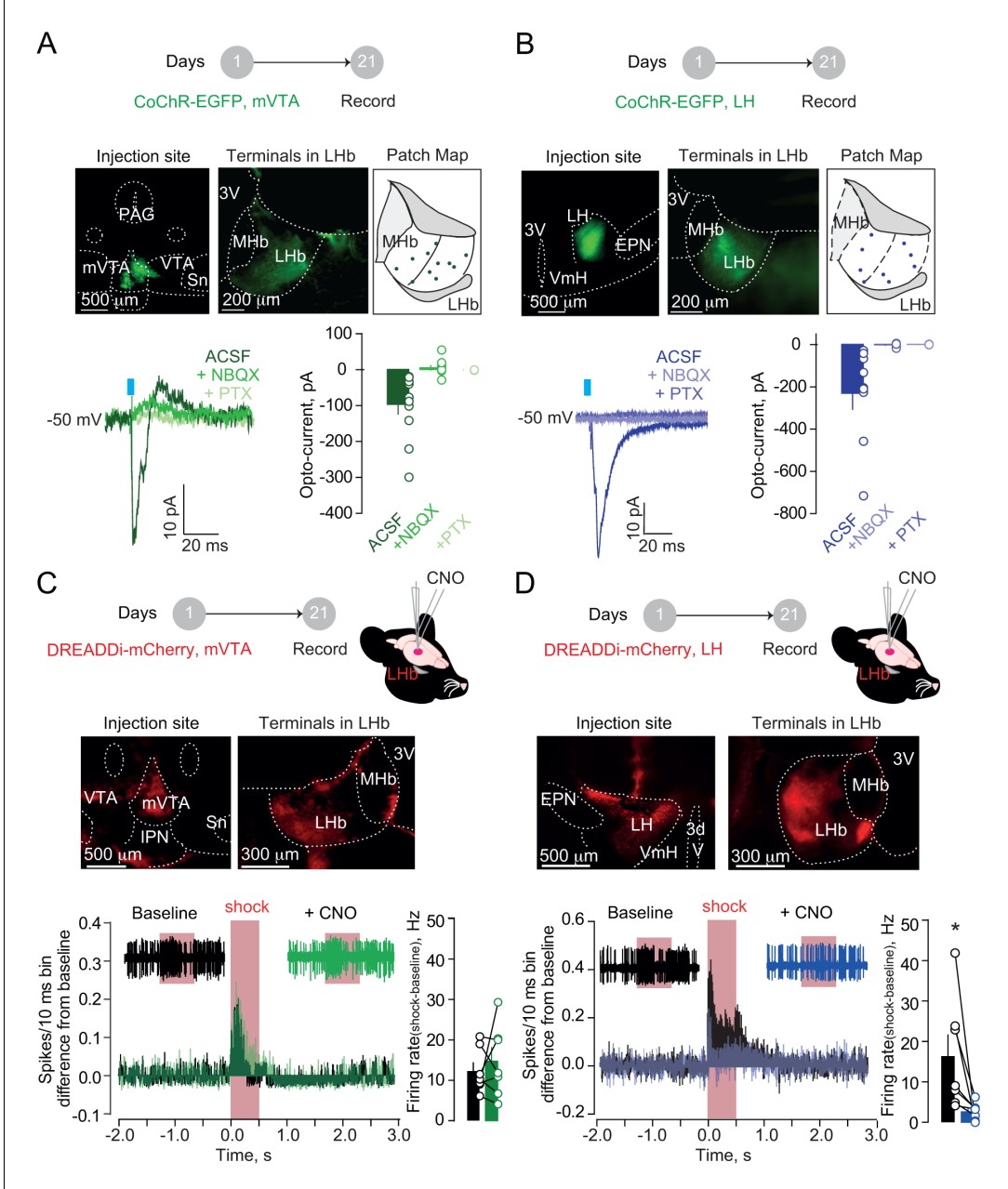

**Figure 2.** Hypothalamic, but not mesencephalic, excitatory projections mediate foot-shock excitation of LHb neurons. (**A**) Experimental timeline, representative images for CoChR expression and recording map in LHb. Bottom. Sample currents and amplitude bar graphs for VTA→LHb terminals optical stimulation at rest (N/n = 4/11). (**B**) Same as (**a**) but for LH→LHb (N/n = 5/9). (**C**) Experimental timeline and DREADDi expression in mVTA somata and LHb terminals. Averaged PSTH, bar graph and scatter plot for Fs-driven excitation before/after local CNO (CNO, 100 μM; N/n = 4/7; paired t-test, t = 0.69 *p*=0.51). (**D**) Same as (**c**) but for LH→LHb projections (N/n = 5/7; paired t-test, t = 2.45 *p*=0.04). Results are reported as mean ± S. E.M. N = mice; n = cells. 3V, third ventricle, MHb, medial habenula, EPN, entopeduncular nucleus, PAG, periaqueductal gray, IPN, interpeduncular nucleus, VTA, ventral tegmental area, Sn, substantia nigra, VmH, ventral medial hypothalamus. See *Figure 2—source data 1*.

DOI: https://doi.org/10.7554/eLife.30697.008

The following source data and figure supplements are available for figure 2:

**Source data 1.**
DOI: https://doi.org/10.7554/eLife.30697.012
**Figure supplement 1.** Activation of hypothalamic and mesencephalic terminals within the LHb triggers place aversion.
DOI: https://doi.org/10.7554/eLife.30697.009
**Figure 2—supplement figure 1—source data 1.**
*Figure 2 continued on next page*

*Figure 2 continued*

DOI: https://doi.org/10.7554/eLife.30697.013

**Figure supplement 2.** DREADDi activation reduces input-specific presynaptic glutamate release but does not affect LHb baseline firing rate in vivo.

DOI: https://doi.org/10.7554/eLife.30697.010

**Figure 2—supplement figure 2—source data 1.**

DOI: https://doi.org/10.7554/eLife.30697.014

**Figure supplement 3.** Neurons in the LH projecting to LHb show fast and phasic excitation upon foot-shock.

DOI: https://doi.org/10.7554/eLife.30697.011

**Figure 2—supplement figure 3—source data 1.**

DOI: https://doi.org/10.7554/eLife.30697.015

threat. To test this, we targeted LH neurons with a Cre-dependent DREADDi construct (rAAV-EF1α-DIO-DREADDi-mCherry). We next infused the canine-derived Cav2-Cre vector within the LHb allowing for retrograde Cre-transport and subsequent DREADDi expression to selectively silence LH→LHb projections in mice (*Figure 4A*). Control (CTRL; rAAV-EF1α-td-Tomato) animals, after CNO injection, successfully shuttled to the compartment opposite to the one of shock delivery with short latencies across trials. Instead, mice expressing DREADDi in LH→LHb escaped with higher latencies, along with increased shuttling failure rate (*Figure 4B*). In contrast, no difference in latencies or failures was detected when silencing VTA→LHb projections consistent with the lack of contribution to foot-shock mediated LHb excitation (*Figure 4—figure supplement 1A*). Importantly, silencing the LH→LHb synapses did not alter locomotion in an open field, nor pain perception, measured with the hot-plate test (*Figure 4—figure supplement 1B*).

Next, we investigated the role of LH→LHb in a more naturalistic setting, mimicking the attack of a predator with a projected shadow to model innate escape (*Kunwar et al., 2015*). When exposing CTRL mice, after CNO injection, to such paradigm, they rapidly escaped from the arena center to a 'safe' area (nest) (*Figure 4C* and *Video 1*). In contrast LH→LHb DREADDi mice had significant larger escape latencies (*Figure 4C* and *Video 2*). Altogether, these data attribute to the LH→LHb projection the crucial role in guiding escape, an evolutionary conserved innate response to a threat.

## Discussion

The functional connectivity allowing hypothalamic circuits to orchestrate defensive and escape behaviors remained to date modestly described (*Kunwar et al., 2015*; *Stamatakis et al., 2016*; *Mongeau et al., 2003*). Our data fit the LHb within this neurocircuitry inferring causality between LH→LHb excitation and the behavioral processing of aversive stimuli. Nevertheless, how sensory information reaches the LH, and how precise aversion-encoding hypothalamic cell-types control LHb remain to be established (*Herrera et al., 2016*; *González et al., 2016*). Aversive-driven escape is in equilibrium with freezing behaviors (innate and learned) that, at least partly, rely on periaqueductal gray and the amygdala (*Fadok et al., 2017*; *Tovote et al., 2016*; *Wei et al., 2015*). This raises interest to decipher whether an anatomical and functional connectivity exists between these latter nuclei and LH→LHb circuits, and whether they encode negative stimuli in synergy or if devoted to specific behavioral aspects (i.e. flight, fight or freezing).

We found that shock-driven LHb excitation and escape behavior are independent of the VTA→LHb projection. However, we report, consistent with published data, that VTA→LHb terminals activation, drives aversive behaviors highlighting that cautious interpretation should be given to motivated behaviors generated by optogenetic manipulation of habenular circuits (*Root et al., 2014a*; *Shabel et al., 2012*; *Stamatakis et al., 2016*; *Yoo et al., 2016*).

Our data suggest a relevant control by LH→LHb neurons onto midbrain GABA neurons, which in turn may lead to aversion-driven dopamine inhibition (*Ungless et al., 2004*). However, we also find that LH→LHb neurons directly synapse onto 5HT and DA neurons. Notably, while habenula lesions reduced reward omission-driven inhibition of DA neurons, it left instead intact the inhibitory response to aversive stimuli (*Tian and Uchida, 2015*). Altogether, this heightens the need of understanding the computational properties and behavioral relevance of cell-type specific LHb projections onto monoaminergic systems (*Lammel et al., 2012*; *Tan et al., 2012*; *van Zessen et al., 2012*; *Schweimer and Ungless, 2010*).

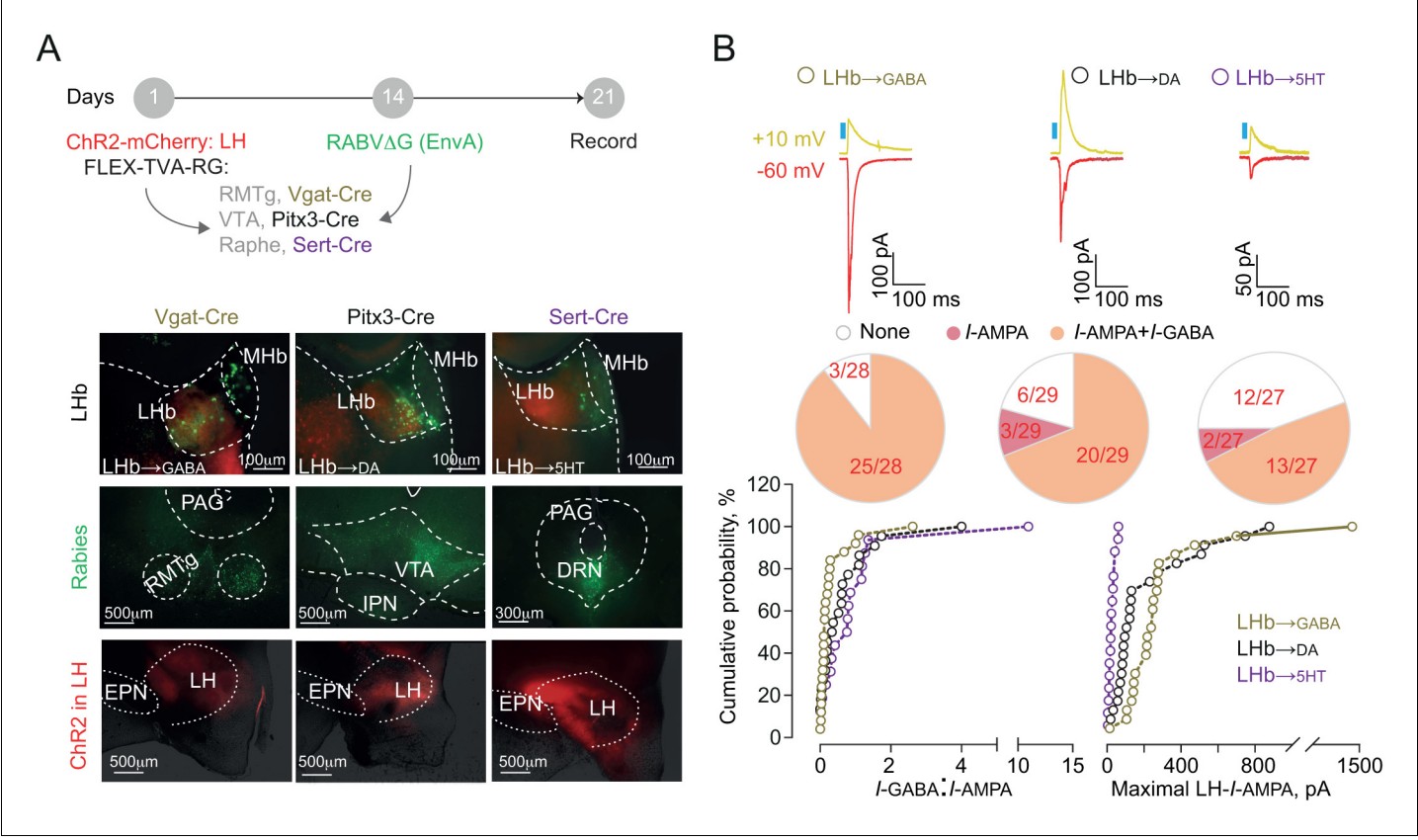

**Figure 3.** Functional output connectivity of hypothalamic-habenular projections. (**A**) Timeline for rabies-based labeling. Bottom. Image illustrating LH-ChR2-mCherry fibers (red) and RABVΔG-(EnvA)-eGFP-retrolabeled LHb neurons (green) projecting to midbrain GABA, DA and 5HT cells. (**B**) Light-evoked glutamatergic/GABAergic currents and connectivity charts for LH→LHb-to-GABA (N/n = 6/28; Connectivity = 89.2%), LH→LHb-to-DA (N/n = 6/29; Connectivity = 79.3%;) and LH→LHb-to-5HT neurons (N/n = 4/27; Connectivity = 55.5%). Bottom. Cumulative probability (Kolmogorov-Smirnov test; *VGat-Cre*, *Pitx3-Cre*, *Sert-Cre*, n = 25, 23, 15) for I-GABA/I-AMPA (*Pitx3* vs *Sert* p=0.33; *VGat* vs *Sert*. ***p=0.0009; *VGat* vs *Pitx3* *p=0.039) and maximal I-AMPA (*VGat* vs *Pitx3* ***p=0.0005; *VGat* vs *Sert* ***p<0.0001; *Pitx3* vs *Sert* ***p<0.0001) recorded. Results are reported as mean ±S.E.M. N = mice; n = cells. MHb, medial habenula, EPN, entopeduncular nucleus, PAG, periaqueductal gray, IPN, interpeduncular nucleus, VTA, ventral tegmental area, RMTg, rostromedial tegmental nucleus, DRN, dorsal raphe nucleus. See *Figure 3—source data 1*.

DOI: https://doi.org/10.7554/eLife.30697.016

The following source data and figure supplements are available for figure 3:

**Source data 1.**

DOI: https://doi.org/10.7554/eLife.30697.018

**Figure supplement 1.** Anatomical and physiological properties of LH-LHb projecting neurons.

DOI: https://doi.org/10.7554/eLife.30697.017

**Figure 3—supplement figure 1—source data 1.**

DOI: https://doi.org/10.7554/eLife.30697.019

In conclusion, our data describe a neural circuit instrumental for escaping a threat, and open in-depth investigation of cell specificities and synaptic adaptations of the LH→LHb projection to better decipher LHb processing in both health and disease states.

# Materials and methods

## Experimental subjects

All in vivo and ex vivo procedures were performed on C57Bl/6J mice (males) wild-type or *slc30a1-Cre* (VGat-Cre), *Pitx3-Cre* and *Slc6a4-Cre* (Sert-Cre) mice aged 4–12 weeks. Mice were used in accordance with the guidelines of the Ministry of Agriculture and Forestry for animal handling and

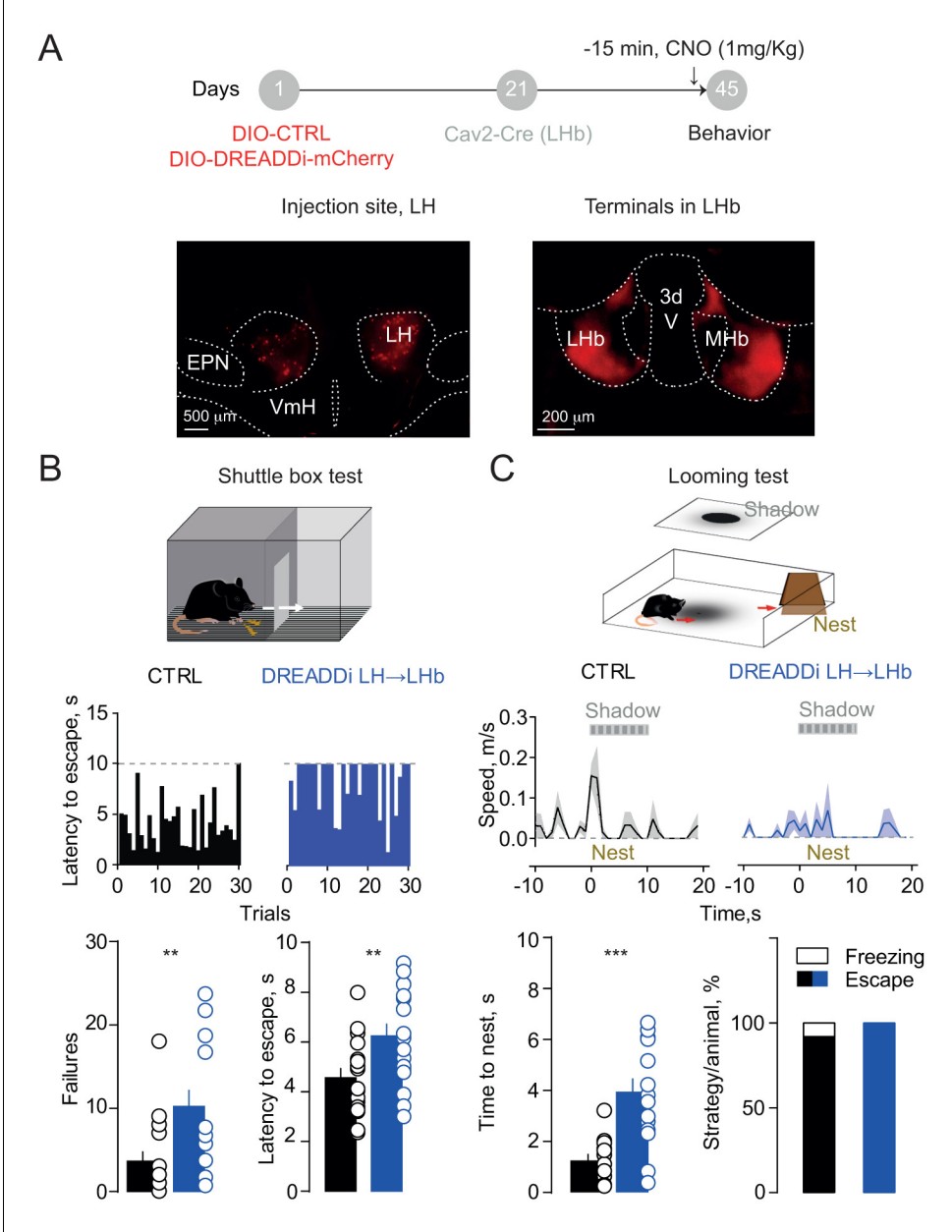

**Figure 4.** Silencing the hypothalamic-habenular projection impairs escape behaviors. (**A**) Timeline for Cre-dependent DREADDi strategy. Image for DREADDi expression (red) in LH somata and LH terminals in LHb. (**B**) Sample histograms for Fs-evoked latency to escape (30 trials) in a CTRL and DREADDi-expressing mouse. Bottom. Bar graphs and scatter plots for failure rates and latencies to escape (CTRL vs DREADDi, N = 17, 17; unpaired t-test; failures: t = 2.97, \*\*p=0.005; latency: t = 2.79, \*\*p=0.008). (**C**) Looming light test. Sample speed across time during the looming test in CTRLs and DREADDi-expressing animals. Bottom. Latency to escape (CTRL vs DREADDi, N = 12, 15; unpaired t-test; t = 4.12, \*\*\*p=0.0004) and strategy adopted upon the looming stimulus (Chi-square test, $X^2$ = 1.29, p=0.25) in the two experimental groups. Results are reported as mean ±S.E.M. N = mice. MHb, medial habenula, EPN, entopeduncular nucleus, VmH, ventral medial hypothalamus. See *Figure 4—source data 1*.

DOI: https://doi.org/10.7554/eLife.30697.020

The following source data and figure supplements are available for figure 4:

**Source data 1.**

DOI: https://doi.org/10.7554/eLife.30697.022

**Figure supplement 1.** Behavioral assessment of DREADDi activation on VTA→LHb and LH→LHb projections.

*Figure 4 continued on next page*

*Figure 4 continued*

DOI: https://doi.org/10.7554/eLife.30697.021

**Figure 4—supplement figure 1—source data 1.**

DOI: https://doi.org/10.7554/eLife.30697.023

the ethic committee Charles Darwin #5 of the University Pierre et Marie Curie. Part of the current study was carried at the Department of Fundamental Neuroscience of the University of Lausanne (Lausanne, Switzerland) according to the regulations of the Cantonal Veterinary Offices of Vaud and Zurich (Switzerland; License VD3171). Mice were housed in groups of 5 per cage with water and food ad libitum. Mice were randomly allocated to experimental groups.

## Stereotactic injections

For surgery, all mice were anaesthetized with ketamine (100 mg/kg)/xylazine (10 mg/kg) (Sigma-Aldrich, France). Viral injections were performed using a glass pipette mounted on a stereotactic frame (Kopf, France). Volumes ranged between 200 and 400 nl per side, infused at a rate of 100–150 nl/min. The injection pipette was withdrawn from the brain 10 min after the infusion.

For in vivo and in vitro electrophysiological experiments we bilaterally injected AAV-CAG-hChR2 (H134R)-mCherry or rAAV2.1-hSyn-CoChr-eGFP (University of North Carolina, US) and rAAV8-Hsyn-Gi-DREADD-mCherry (University of Pennsylvania, US), either alone or in a mixture.

C57B6J mice (4–7 weeks) were injected in the Lateral hypothalamus (LH, −1.35 mm AP, 0.9 mm ML, −5.2 mm DV) or in the medial portion of the ventral tegmental area (mVTA, −2.6 mm AP, 0.0 mm ML, −4.8 mm DV). Animals were allowed to recover for a minimum of three weeks before recordings were performed.

For in vivo optogenetic experiments we injected mice in the LH or mVTA with rAAV2.1-hSyn-CoChr-eGFP (University of North Carolina, US) or rAAV2.1-CAG-dt-Tomato as a control virus. A single fiber optic was placed above and medially to the LHb (−1.4 mm AP, 0.2 mm ML, 2.8 mm DV) and then cemented on the mouse skull (Superbond resin cements, Sun medical, Japan).

In a separate set of experiments mice were injected unilaterally in the LHb (−1.4 mm AP, 0.45 ML, 3.1 mm DV) with the genetically encoded calcium indicator GCamp6f (rAAV2.1- hSyn-GCamp6f-eGFP; Pennsylvania University, US). A single fiber probe/optical fiber was placed and fixed 0.5 mm above the injection site. Only mice expressing GCamp6f-eGFP were used for such experiments.

For opto-tagging experiments in the LH, a herpesvirus carrying channelrhodopsin-2 (HSV-hEF1α-hChR2(H134R)-mCherry-HT; McGovern Institute of Brain Research, MIT, US) was injected unilaterally in the LHb. In vivo recordings were performed at the coordinates of the ipsilateral LH (−1.0 ÷ −1.4 mm AP; 0.8 ÷ 1.0 mm ML; 4.8 ÷ 5.3 mm DV from the bregma) 2 weeks after recovery.

For anatomical tracing studies *Pitx3*-Cre,*VGAT*-ires-Cre and *Sert*-CRE mice (4–7 weeks) were injected respectively with rabies helper virus rAAV1/2-Flex-TVA-RG (titer: $1 \times 10^{12}$ pp/ml) in the VTA (−2.2 mm AP, 0.65 mm ML, −4.6 mm DV) or RMTg (−2.9 mm AP, 0.5 mm ML, −4.3 mm DV) or dorsal raphe (−3.1 mm AP; 0 mm ML; −4.3 mm DV from the skull surface). Animals were allowed to recover for two weeks, then re-injected at the same coordinates with modified glycoprotein-deleted pseudotyped rabies-EGFP (RABVΔG (EnvA)-EGFP; titer: $1 \times 10^7$ pp/ml).

For behavioral studies, C57B6J mice were infused with rAAV8-hSyn-DIO-HM4Di-mCherry virus (University of Pennsylvania, US) in the LH or mVTA. After a delay period of 3–5 weeks, mice were injected in the LHb (−1.45 mm AP, 0.45 mm ML, −3 mm DV) with retrograde Cav2-Cre virus (titer: ~2,5 $\times$ $10^{12}$ pp/ml, IGMM CNRS, France). The injection sites were examined for all experiments and only data from animals with correct injections were included in the analysis.

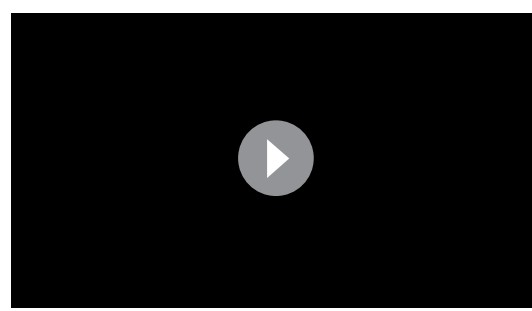

**Video 1.** Escape behavior in mice. Looming test performance of a representative mouse infused with a control virus within the LH and Cav2-Cre in the LHb.
DOI: https://doi.org/10.7554/eLife.30697.024

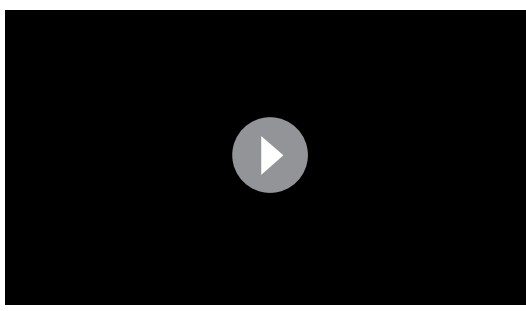

**Video 2.** Silencing the LH-to-LHb pathway compromise escape behavior. Looming test performance of a representative mouse infused with a AAV-hSyn-DIO-hM4Di-mCherry within the LH and Cav2-Cre in the LHb. DOI: https://doi.org/10.7554/eLife.30697.025

## Real-time place aversion and optogenetics

Four weeks following surgery, mice implanted with optical fibers above the LHb were placed in a custom-made behavioral arena with two compartments with different visual cues and wall texture for 15 min. The counterbalanced side of the chamber was defined as the light-paired stimulation side. At the beginning of the session, the mouse was placed in the non-light paired side of the chamber. Every time the mouse crossed to the light-paired side a 20 Hz constant laser stimulation (473 nm, ~10 mW) was delivered. Light stimulation was interrupted when the mouse exit the light-paired side. Time spent on the stimulation-paired side and velocity were recorded via a digital camera interfaced with Any-maze software (Stoelting, Ireland).

## In vivo photometry experiments

Fiber photometry experiments were performed using a custom-made set-up (airpuff experiments, Burdakov lab, London) and a photometry system using time-correlated photon counting technology (TCSPC) (footshocks experiments, ChiSquare Bioimaging, US)(*Cui et al., 2014*).

Airpuffs experiments. Two fiber-coupled LEDs (Thorlabs M470F3 and M4051FP1) provided excitation light at different wavelengths: 470 nm to excite the calcium-dependent range of GCaMP6f, and 405 nm, the isosbestic excitation wavelength of GCaMP6f. Square pulses of 470 nm and 405 nm excitation light were alternated at 20 Hz, allowing near-simultaneous recording of, respectively, calcium-related fluorescence changes and a reference signal reporting calcium-independent fluorescence changes (e.g. movement artefacts; *Kim et al., 2016*). The light paths of the two excitation wavelengths were aligned using a custom-built fiber-coupled filter cube containing a dichroic mirror (Thorlabs DMLP425R), preceded by a collimator (350–700 nm; Thorlabs F240FC-A) and GFP excitation filter (Thorlabs MF469-35) for the 470 light path, and a collimator (395–415 nm; Thorlabs F671FC-405) for the 405 light path. Output excitation light was fiber-coupled to a Doric Fluorescence Mini Cube (FMC3_E(460-490)_F(500-550)), with the excitation filter removed. A single multimode patch cable was used for excitation and emission inside the brain (Thorlabs, 200 μm core, 0.39 NA), connected to the Doric cube using a metal FC connector and plugged into the fiber-optic implant using a metal ferrule-to-ferrule connector. Emitted fluorescence (~525 nm, regardless of excitation light) was filtered at the Doric cube, and then detected and amplified by a fiber-coupled photodetector (Thorlabs PDF10A). The light power of the 470 nm and 405 nm excitation wavelengths were set separately according to the signal seen in each mouse, and measured after the mouse was unplugged (range: 40–100 μW). The analogue signal from the photodetector was sent to an ADC input of a CED Micro1401, and signals were recorded using Spike2. Offline, 470- and 405-excited signals were separated according to the time of excitation light, and $\Delta F/F$ was calculated for each, according to $(F-F_0)/F_0$, where $F_0$ was defined as the median of a 10 s baseline period (for the air puff experiment, the first 10 s of each trial). $\Delta F/F_{405}$ was subtracted from $\Delta F/F_{470}$, to remove any non-calcium-related fluorescence changes from the signal.

Puffs of compressed air lasting approximately 1 s were delivered to the base of the tail, every 60 s. This was done 5–7 times per mouse, and was aversive (mice moved quickly away) but not painful.

Foot-shocks experiments: Footshocks (3 s) were delivered in a chamber provided with an electrified grid-floor with an intensity of 0.3mA. Each mouse tested received 5–10 shocks in total with an inter-shock interval of 60 s. The mouse immediately reacted to the shock escaping in the other side of the chamber. A subset of animals was tested in the rotarod while monitoring locomotion. RPM increased continuously from 3 to 30 in bouts of 3 RPMs. The experiments were replicated two to three times in the laboratory.

## Electrophysiology

### In vivo recordings

Mice were anesthetized with isoflurane (Univentor, Malta. Induction: 2%; maintenance: 1–1.5%), and placed in the stereotaxic apparatus (Kopf, Germany). Their body temperature was maintained at 36 ± 1 °C using a feedback-controlled heating pad (CMA 450 Temperature Controller, USA). The scalp was retracted and one burr hole was drilled above the LHb (AP: −1.3 − −1.6 mm, L: 0.4–0.5 mm, V: −2.3 − −3.2) for the placement of a recording electrode. Single unit activity was recorded extracellularly using glass micropipettes filled with 2% pontamine sky blue dissolved in 0.5 M sodium acetate (impedance 3–6 MΩ). Signal was filtered (band-pass 500–5000 Hz), pre-amplified (DAM80, WPI, Germany), amplified and (Neurolog System, Digitimer, UK), displayed on a digital storage oscilloscope (OX 530, Metrix, USA).

Experiments were sampled on- and off-line by a computer connected to CED Power 1401 laboratory interface (Cambridge Electronic Design, Cambridge, UK) running the Spike2 software (Cambridge Electronic Design). Single units were isolated and the spontaneous activity was recorded for 5 min before triggering the shock protocol.

Spontaneous firing rate, percent of spikes in bursts and coefficient of variation (CV = standard deviation of interspike intervals/mean interspike interval; a measure of firing regularity) were determined. The criteria to identify a burst were made by a qualitative analysis per each neuron of the interspike interval histogram of 200 s of recordings. We defined the initiation of a burst as at least 2 action potentials occurred in an interval <10 ms. The burst was considered finished when the interval between the last 2 action potentials was >20 ms. Additionally, autocorrelograms were generated using a 10 ms bin width for intervals up to 2 s, to qualitatively classify neurons as firing in the regular, irregular or burst firing mode. Autocorrelograms showing three or more regularly occurring peaks were characteristic of the regular firing pattern. An initial trough that rose smoothly to a steady state was classified as irregular firing pattern, whereas an initial peak, followed by decay to a steady state, was indicating a burst pattern the bursting mode (elsewhere (aptic physiology with Prof. Fernando Valenzuela at the University of New Mexico, of the project(*Lecca et al., 2011*).

After recording baseline activity, each cell was tested for its response to repetitive (each 5 s) shocks (0.5 s, 3mA) delivered to the hind paw contralateral to the recording side with a spike2 automatic program. PSTHs and raster plots were built from 20 to 30 shocks and displayed using 10 ms bin width. A cell was considered excited when the mean number of action potentials/bin (bin length = 10 ms) in at least one of the four epochs (50 ms *per* epoch) after the shock inset was 2 times the Standard Deviation (SD) higher than baseline levels (the average number of action potentials/bin in the 2 s period before the shock). The onset of the response was calculated as the first of at least 2 consecutive bin higher than the 2SD of the averaged baseline. The duration of the response was calculated from the latency to the first of at least 5 consecutive bins not different than the baseline +2 SD. The magnitude of the response was obtained subtracting the baseline firing rate to the firing during the duration of the shock response.

Graphical representation of the foot-shocks responses were obtain as follow: for each cell excited by foot-shocks we normalized the PSTH subtracting the averaged baseline (2 s prior the shock, bin 10 ms) to every bin divided the number of sweeps. Then we implemented all the cells in a single histogram graph, reporting the mean ± the s.E.M.

For the recordings in the lateral hypothalamus (AP, −1.0 − −1.4 mm; L, 0.8–1.0 mm) we lowered the recording pipette at the following depth: V, from –4.7 to – 5.2 mm. In this case to select LHb-projecting LH neurons, we tested each cell for its response to the ChR2-activating light. Only opto-identified neurons were considered for the analysis. Firing rate, coefficient of variation (%) as well as the response to foot-shocks were assessed using the same criteria for the LHb neurons.

A double barrel pipette assembly (injection tip,<50 μm in diameter attached ~100 μm beyond the recording tip) was used for recording LHb spike activity with simultaneous local microinjection of drugs. The injection pipette was filled with one of the following: a mixture of the specific NMDA antagonist amino-5-phosphonopentanoic acid (AP-5; 100 μm) and the specific non-NMDA 2,3-Dioxo-6-nitro-1,2,3,4-tetrahydrobenzo[f]quinoxaline-7-sulfonamide (NBQX, 100 μM), or phosphate buffered saline. CNO (100 μM) was used for local chemogenetic silencing. Drugs were microinjected into the LHb, using brief pulses of pneumatic pressure (40 psi, 40 ms, Picospritzer, IM-300,

Narishige, Japan). In all experiments a total volume of 60 nl was infused over 30 s for each injection. Two injections maximum *per* animal were given at an interval >30 min.

At the end of each experiment, the electrode placement was determined with an iontophoretic deposit of pontamine sky blue dye (−80 µA, continuous current for 5 min). Brains were then rapidly removed and fixed in 4% paraformaldehyde solution. The position of the electrodes was microscopically identified on serial sections (60 µm). Only recordings in the correct area were considered for analysis.

### In vitro recordings

Animals were anesthetized with Ketamine/Xylazine (100 mg/10 mg $Kg^{-1}$ i.p.; Sigma-Aldrich, France).

The preparation of LHb-containing brain slices was done in bubbled ice-cold 95% O2/5% CO2-equilibrated solution containing (in mM): cholineCl 110; glucose 25; NaHCO3 25; MgCl2 7; ascorbic acid 11.6; Na+-pyruvate 3.1; KCl 2.5; NaH2PO4 1.25; CaCl2 0.5. Coronal slices (250 µm) were stored at room temperature in 95% O2/5% CO2-equilibrated artificial cerebrospinal fluid (ACSF) containing (in mM): NaCl 124; NaHCO3 26.2; glucose 11; KCl 2.5; CaCl2 2.5; MgCl2 1.3; NaH2PO4 1. Recordings (flow rate of 2.5 ml/min) were made under an Olympus-BX51 microscope (Olympus, France) at 32°C. Currents were amplified, filtered at 5 kHz and digitized at 20 kHz. Access resistance was monitored by a step of –4 mV (0.1 Hz). Experiments were discarded if the access resistance increased more than 20%.

The internal solution to measure excitatory and inhibitory currents contained (in mM): Cs-methanesulphonate 120, CsCl 10, HEPES 10, EGTA 10, creatine phosphate 5; Na2ATP 4; Na3GTP 0.4, with a liquid junction potential of −16 mV. Under these conditions, it was possible to detect inward glutamatergic currents (at −50/60 mV) and outward GABAergic (at +10 mV). When clamping neurons at −50/60 mV, the inward current was predominantly AMPAR-mediated (blocked by NBQX, 20 µM). In contrast, when clamping neurons at +10 mV, outward currents were predominantly GABAaR-mediated (blocked by picrotoxin, 100 µM).

## Behavioral paradigms

All behavioral tests were conducted during the light phase (7:00–19:00). Mice were habituated to the experimental room light level (35 lux) in their home cage (5 mice/cage) for at least 1 hr prior the testing. Animals were used for maximum two behavioral paradigms compatible one another, with behavioral paradigms repeated at least twice. Animals were randomly assigned to the experimental groups. Operators were blind to the experimental group during the scoring. All mice used for DREADDi silencing experiments received a single injection of CNO (1 mg/kg i.p.) ~20 min prior the test.

## Hot plate test

A standard hot plate (Biosed, Chaville, France), adjusted to 52°C, was used to assess motor reactions in response to noxious stimuli. Mice were confined on the plate by a Plexiglas cylinder (diameter 19 cm, height 26 cm). The latency to a hind paw response (licking or shaking) or jumping was taken as the nociceptive threshold.

## Locomotor activity

To assess the locomotor activity we tested mice in an open field arena. Mice were placed in the center of a plastic box (50 cm x 50 cm x 45 cm) in a room with dim light. Following a 5 min habituation period, the animal's behavior was videotaped and subsequently analyzed (Any-maze, France).

## Shuttle box test

A shuttle box (13 cm × 18 cm × 30 cm) was equipped with an electrified grid floor and a door separating the two compartments. The test session consisted of 30 trials of escapable foot-shocks (10 s at 0.1–0.3 mA) separated by an interval of 30 s. The shock ended when animals shuttled to the opposite compartment. Failure was defined as the absence of shuttling to the other compartment within the 10 s of shock delivery. The time employed by the mouse to shuttle in the other compartment during the shock (latency) was also calculated (*Lecca et al., 2016*).

## Looming visual stimulus test

Mice were tested for behavior in a looming visual stimulus test, as described elsewhere (*Yilmaz and Meister, 2013*). Animals were placed in an open-top Plexiglas box (50 × 50 × 45 cm). A triangular shaped nest (20 × 12 cm) was placed in one corner. Recordings were performed under illumination provided by the projector screen placed above the arena. After 10 min of habituation, a looming stimulus was presented from the screen when an animal was in the center. The stimulus of 0.5 s duration was repeated 10 times with an interstimulus interval of 0.5 s. The latency to escape and the freezing time after escaping was calculated for each mouse. The analysis was performed off-line (Anymaze, Ireland).

## Drugs

The amino-5-phosphonopentanoic acid (AP-5), the specific non-NMDA 2,3-Dioxo-6-nitro-1,2,3,4-tetrahydrobenzo[f]quinoxaline-7-sulfonamide (NBQX), the GABAA antagonist (picrotoxin) and the clozapine N-oxide (CNO) were obtained from Abcam (UK) and Tocris (UK). With the exception of picrotoxin and CNO (DMSO, 0.01% final concentration), all drugs were dissolved in purified water.

## Statistical analysis

Online/offline analyses were performed using Spike2 (Cambridge Electronic Design) IGOR-6 (Wavemetrics, US) and Prism (Graphpad, US). Data distribution was systematically tested with D'Agostino Pearson and Shapiro-Wilk normality tests. Depending on the distribution, parametric or not parametric test were used. Single data points are always plotted. Electrophysiological and behavioral experiments were replicated at least three times within the laboratory. Sample size was pre-estimated from previously published research and from pilot experiments performed in the laboratory. Compiled data are expressed as mean ± S.E.M. Significance was set at $p < 0.05$ using two-sided unpaired t-test, Kolmogorov-Smirnov test, one or two-way ANOVA with multiple comparison when applicable. The use of the paired t-test and two way ANOVA for repeated measured were stated in the legend figure text. The Chi-Square test was used when required.

## Acknowledgements

This work was supported by INSERM Atip-Avenir, the City of Paris, the European Research Council (Starting grant SalienSy 335333) to MM, the HFSP (Young Investigator Award RGY0076) to DB. We thank A Adamantidis, J Letzkus, C Lüscher and the entire Mameli Laboratory for feedback on the manuscript and constructive discussions.

## Additional information

### Funding

| Funder | Grant reference number | Author |
|---|---|---|
| Nederlandse Organisatie voor Wetenschappelijk Onderzoek | VENI Grant 863.15.012 | Frank Julius Meye |
| Brain and Behavior Research Foundation | NARSAD Young Investigator Grant 25190 | Frank Julius Meye |
| Human Frontier Science Program | Young Investigator Award RGY0076 | Denis Burdakov |
| European Research Council | Starting grant SalienSy 335333 | Manuel Mameli |

The funders had no role in study design, data collection and interpretation, or the decision to submit the work for publication.

### Author contributions

Salvatore Lecca, Conceptualization, Data curation, Formal analysis, Validation, Writing—original draft, Writing—review and editing; Frank Julius Meye, Massimo Trusel, Anna Tchenio, Julia Harris,

Data curation, Formal analysis; Martin Karl Schwarz, Resources; Denis Burdakov, Resources, Supervision; Francois Georges, Supervision; Manuel Mameli, Conceptualization, Resources, Supervision, Funding acquisition, Methodology, Writing—original draft, Project administration, Writing—review and editing

## Author ORCIDs
Manuel Mameli http://orcid.org/0000-0002-0570-6964

## Ethics

Animal experimentation: Mice were used in accordance with the guidelines of the Ministry of Agriculture and Forestry for animal handling and the ethic committee Charles Darwin #5 of the University Pierre et Marie Curie. Part of the current study was carried at the Department of Fundamental Neuroscience of the University of Lausanne (Lausanne, Switzerland) according to the regulations of the Cantonal Veterinary Offices of Vaud and Zurich (Switzerland; License VD3171).

## Decision letter and Author response

Decision letter https://doi.org/10.7554/eLife.30697.027
Author response https://doi.org/10.7554/eLife.30697.028

---

# Additional files

## Supplementary files
• Transparent reporting form
DOI: https://doi.org/10.7554/eLife.30697.026

---

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
