## [Decision Letter]

Thank you for submitting your article "Aversive stimuli drive hypothalamus-to-habenula excitation to promote escape behavior" for consideration by *eLife*. Your article has been favorably evaluated by Gary Westbrook (Senior Editor) and three reviewers, one of whom, Olivier Manzoni (Reviewer #1), is a member of our Board of Reviewing Editors. The following individuals involved in review of your submission have agreed to reveal their identity: Stephan Lammel (Reviewer #2); Garret D Stuber (Reviewer #3).

The reviewers have discussed the reviews with one another and the Reviewing Editor has drafted this decision to help you prepare a revised submission. Your revisions should focus on the Summary, Essential revisions and Minor points. The text of the original reviews before the discussion are included below for your information.

Summary:

This manuscript describes how aversive stimuli modulate the lateral hypothalamus (LH) to Lateral Habenula (LHb) circuitry to promote escape behavior. Fiber photometry and in vivo electrophysiology were used to show that aversive stimuli (foot shock and air puffs) increase excitation in a subset of LHb neurons. Using a chemogenetic approach the data show that LH, but not medial ventral tegmental area, inputs to the LHb are involved in foot-shock excitation of LHb neurons. Furthermore, trans-synaptic rabies virus tracing revealed the functional output connectivity of LH to LHb input and demonstrate functional connectivity between LH neurons and LHb neurons that synapse onto midbrain GABA neurons. Finally, selective chemogenetic silencing of LH inputs to the LHb impairs aversion-driven escape behaviors. Altogether these results shed light on the behavioral relevance and function of excitatory synapses from the Lateral Hypothalamus to the Lateral Habenula in driving escape behavior in mice.

Essential revisions:

1) A major concern is whether increases in activity in the LH-LHb pathway is tightly correlated with locomotion. Presumably there is increased movement while the mice are escaping from foot-shock. Does activity in this pathway occur at similar movement velocities in the absence of foot-shock?

2) The discussion of the previous literature should be better integrated to the manuscript in order to help provide a better rationale for your study. Specifically, the papers by Lammel and Malenka., Nature 2013, Stamatakis and Stuber, Nature Neuroscience, 2012, as well as papers from both Bo Li's and Roberto Malinow's lab where nearby inputs from the basal ganglia to the LHb and their potentiation following foot-shock exposure have been studied must be cited in the Introduction.

3) Please show higher resolution and/or higher magnification images for the fluorescent images (e.g., the images that show terminal expression in Figure 2 and the results from the transsynaptic rabies virus tracing experiments in Figure 3).

4) It is unclear if Figure 1 is from a representative animal or if this is average data from the cohort of mice tested. This should be clarified, and if possible, the average data across trials from all mice included.

5) It is unclear if fluorophore-only expressing controls were used for comparison purposes in the photometry experiments. Please clarify.

Minor points:

1) Gcamp should be changed to GCaMP6f throughout the manuscript.

2) Although some of the data may be non-normally distributed, it is unclear whether all data were tested for normality. Whether it was the case and if so, if the appropriate statistical tests were chosen must be clearly explained.

3) Please check the spelling throughout the manuscript (e.g. subsection “Stereotactic injections”).

*Reviewer #1:*

The well-described results, mixing ex-vivo/in-vivo electrophysiology and behavioral methods shed light on the habenular neurocircuitry encoding a physiological and escape response to external ethologically relevant threats: LH neurons -> LHb neurons connecting midbrain GABA neurons. The methods are state of the art and the experiments astutely designed. One could only regret that this study does not include the exploration of this very circuit in models of chronic stress and/or depression.

*Reviewer #1 (Minor Comments):*

The conceptual advance of this study is somewhat debatable, to the extent that the role of LHb in the response to stressors and aversive behaviors/depression is well established.

Spelling must be checked. e.g. subsection “Stereotactic injections”.

*Reviewer #2:*

This manuscript by Lecca et al. describes an interesting study on how aversive stimuli modulate LH to LHb circuitry to promote escape behavior. First, using fiber photometry and in vivo electrophysiology the authors show that aversive stimuli (foot shock and air puffs) increase excitation in a subset of LHb neurons. Next, they investigate the functional connectivity of two prominent afferent inputs to the LHb (mVTA and LH). Using a chemogenetic approach they were able to demonstrate that LH, but not mVTA, inputs to the LHb are involved in foot-shock excitation of LHb neurons. In addition, using trans-synaptic rabies virus tracing they further examine the functional output connectivity of LH to LHb input and demonstrate functional connectivity between LH neurons and LHb neurons that synapse onto midbrain GABA neurons. Finally, they show that selective chemogenetic silencing of LH inputs to the LHb impairs aversion-driven escape behaviors.

Altogether this is a very nice set of results implicating a specific role for LH inputs to the LHb in escape behavior. The experiments are performed well and described succinctly and add a tremendous amount of new and informative data that will add significantly to our understanding of LHb circuitry and its function. I only have a minor suggestion that would add additional clarity:

The resolution of some of the fluorescence images seems to be very low (e.g., the images that show terminal expression in Figure 2 and the results from the transsynaptic rabies virus tracing experiments in Figure 3). It would be great if the authors can show higher resolution and/or higher magnification images.

*Reviewer #3:*

In this manuscript by Lecca et al., the authors study the behavioral relevance and function of excitatory synapses from the Lateral Hypothalamus to the Lateral Habenula in driving escape behavior in mice. Using slice electrophysiology, opto and chemogenetics and rabies tracing the authors clearly demonstrate a role of the LH-LHb circuit in playing a prominent role in driving escape behavior in response to aversive footshocks or airpuffs. The study builds on a number of previous studies which have also implicated this pathway in driving behavioral aversion. Overall, the paper is clearly written, and the experiments are well-conducted. The presentation of the data is very clear and including the raw data in the manuscript is helpful. I have a few relatively minor concerns for the authors to consider.

1) The discussion of the previous literature could use a bit of work, especially in the Introduction. There it seems to be a few oversights that if included, would help provide additional rationale for the current study. Some of these papers are cited in other parts of the manuscript, but they deserve additional attention in the Introduction. These are papers by Lammel and Malenka., Nature 2013, Stamatakis and Stuber, NN, 2012, Papers by Bo Li's lab and Roberto Malinow's lab which have studied nearby inputs from the basal ganglia to the LHb and their potentiation following foot-shock exposure.

2) Figure 1: unclear if this graph is from a representative animal or if this is average data from the cohort of mice tested. This should be clarified, and if possible, the average data across trials from all mice should be included.

3) It is unclear of Fluorophore only expressing controls where used for comparison purposes in the photometry experiments.

4) A major concern is whether increases in activity in the LH-LHb pathway is tightly correlated with locomotion. Presumable there is increased movement while the mice are escaping from foot-shock. Does activity in this pathway occur at similar movement velocities in the absence of foot-shock?

*Reviewer #3 (Minor Comments):*

1) Gcamp should be changed to GCaMP6f throughout the manuscript.

2) Some of the data may be non-normally distributed. It is unclear whether all data were tested for normality and if so, if the appropriate statistical tests were chosen.

---

## [Author Response]

Essential revisions:1) A major concern is whether increases in activity in the LH-LHb pathway is tightly correlated with locomotion. Presumably there is increased movement while the mice are escaping from foot-shock. Does activity in this pathway occur at similar movement velocities in the absence of foot-shock?

We thank the reviewer for raising this point. We performed a new set of experiments where implanted GCamp6f-expressing mice (in LHb) were initially tested for Foot-shock escape, and in a second step in the rotarod. The latter allows to measure neuronal activity while keeping a tight control on the speed of locomotion in absence of motivational trigger.

After a short baseline, RPM ramped up over time. We find that fluorescent signal did not increase along with speed, remaining rather constant (Figure 1—figure supplement 1).

Notably, the speed at the max RPM was slightly lower than the one promoted by shock due to the fact that mice would fall from the apparatus at such velocities.

These data reported in the revised Figure 1—figure supplement 1 indicate that passive speed increase seems not to engage activity of the LHb. However, it still remains unclear whether LHb activity and speed are interconnected when the motivational component is onboard. This might be plausible in light of the motor-related downstream circuit connectivity of LHb (i.e. SNc). We feel that such topic falls out of the scope of our work, but it is an important question to address promptly.

2) The discussion of the previous literature should be better integrated to the manuscript in order to help provide a better rationale for your study. Specifically, the papers by Lammel and Malenka., Nature 2013, Stamatakis and Stuber, Nature Neuroscience, 2012, as well as papers from both Bo Li's and Roberto Malinow's lab where nearby inputs from the basal ganglia to the LHb and their potentiation following foot-shock exposure have been studied must be cited in the Introduction.

Done.

3) Please show higher resolution and/or higher magnification images for the fluorescent images (e.g., the images that show terminal expression in Figure 2 and the results from the transsynaptic rabies virus tracing experiments in Figure 3).

We agree with the reviewer that the previously uploaded version of the manuscript contained blurry images. We believe this was due to format conversion. We ameliorated the original images, and now uploaded high quality figures where contrast and sharpness has improved. However, we would like to stress the fact that such images were not obtained in thin slices under a confocal microscope. Given that they are associated to slice recordings, such images are from 250μm-thick slices obtained from a regular fluorescent microscope.

4) It is unclear if Figure 1 is from a representative animal or if this is average data from the cohort of mice tested. This should be clarified, and if possible, the average data across trials from all mice included.

We apologize with the reviewer as we indeed did not clarify this in the previous manuscript version. We have now modified Figure 1 implementing the average time line of latency obtained from the entire cohort of mice.

5) It is unclear if fluorophore-only expressing controls were used for comparison purposes in the photometry experiments. Please clarify.

The reviewer raises an interesting technical issue. We did not implant a cohort of mice with AAV-fluorophore only. We determined neuronal responses only in comparison with baseline state, as our protocol allows time-locked analysis with behavior. Furthermore, the new data obtained in the rotarod suggest that fluorescent signal is relatively stable over time.

Minor points:1) Gcamp should be changed to GCaMP6f throughout the manuscript.

Done.

2) Although some of the data may be non-normally distributed, it is unclear whether all data were tested for normality. Whether it was the case and if so, if the appropriate statistical tests were chosen must be clearly explained.

We apologize if this was unclear. In the statistical analysis paragraph of our Materials and methods section we stated that normality was systematically tested (D’Agostino Pearson). We re-tested for normal distribution implementing the Shapiro-Wilk normality test. We confirmed the use of the statistical analysis previously employed. This is indicated by the different tests specifically stated in each figure legend.

3) Please check the spelling throughout the manuscript (e.g. subsection “Stereotactic injections”).

Done.